# Oxidation Protection of High-Temperature Coatings on the Surface of Mo-Based Alloys—A Review

**Tao Fu, Fuqiang Shen, Yingyi Zhang \*, Laihao Yu, Kunkun Cui, Jie Wang and Xu Zhang**

School of Metallurgical Engineering, Anhui University of Technology, Maanshan 243002, China;
ahgydxtaofu@163.com (T.F.); sfq19556630201@126.com (F.S.); aa1120407@126.com (L.Y.);
15613581810@163.com (K.C.); wangjiemaster0101@outlook.com (J.W.); zx13013111171@163.com (X.Z.)
\* Correspondence: zhangyingyi@cqu.edu.cn

**Abstract:** Molybdenum and its alloys, with high melting points, excellent corrosion resistance and high temperature creep resistance, are a vital high-temperature structural material. However, the poor oxidation resistance at high temperatures is a major barrier to their application. This work provides a summary of surface modification techniques for Mo and its alloys under high-temperature aerobic conditions of nearly half a century, including slurry sintering technology, plasma spraying technology, chemical vapor deposition technology, and liquid phase deposition technology. The microstructure and oxidation behavior of various coatings were analyzed. The advantages and disadvantages of various processes were compared, and the key measures to improve oxidation resistance of coatings were also outlined. The future research direction in this field is set out.

**Keywords:** molybdenum alloys; coating; oxidation behavior; microstructure; high-temperature

## 1. Introduction

With the rapid development of aerospace, national defense and the military industry, electronics, and so on, increasing attention has been paid to the research and application of refractory metals [1–4]. Molybdenum and molybdenum-based alloys have a high melting point (2620 °C), good high-temperature mechanical properties and high conductivity and thermal conductivity, and are widely used in high-temperature structures [5–10]. However, the alloys have a poor oxidation resistance, and the "Pesting oxidation" at 400–800 °C and oxidation decomposition above 1000 °C are the main factors that limit their application [11–14]. At present, the alloying and surface-coating technology are the main methods to increase the oxidation resistance of the basal materials [15,16]. The types of molybdenum alloys and the various surface coating technologies of Mo and its alloys are shown in Figure 1 [17–20]. It can been seen that the Ti, Zr, W, Re, Si, B, Hf, C and rare earth oxides are often added to pure Mo as beneficial elements to prepare molybdenum alloys. However, the result of alloying is not satisfactory when considering the mechanical properties and high-temperature oxidation resistance of the alloys [21,22]. For example, adding a certain amount Ti element to the alloy can enhance its strength, but it will further accelerate the oxidation of the alloy [23]. Mo–Si–B alloys have satisfactory high temperature oxidation resistance, but their fracture toughness is poor. Mo–Ti–Si–B alloys are considered as a promising ultra-high temperature material. However, their oxidation resistance and mechanical properties need to be further studied [24]. In contrast, the surface-coating technology can improve the oxidation resistance of the alloy at high temperature with as little impact on the mechanical properties as possible. Therefore, it is favored by the majority of researchers [25].

In past work, we discussed the composition, structure and oxidation characteristics of HAPC coating on the surface of molybdenum and its alloys in detail [26]. However, there are almost no reviews reporting on research about other methods in this field [27]. In this work, the latest research progress of high-temperature oxidation resistance coatings on the

surface of molybdenum and its alloys is reviewed. The characteristics of different surface-coating preparation technologies are summarized and analyzed, including slurry sintering, plasma spraying, chemical vapor deposition, and liquid-phase deposition. As an important physical vapor deposition technology, magnetron sputtering technology is also widely used in metal surface coating [28]. In addition, the molten salt and laser cladding technologies are also mentioned [29–39]. The composition, structure and oxidation characteristics of all kinds of coatings have been given in relevant figures and tables [40]. Moreover, the process characteristics of various methods and key measures to improve the oxidation resistance of coatings are pointed out. The future research and development direction in this field will be outlined.

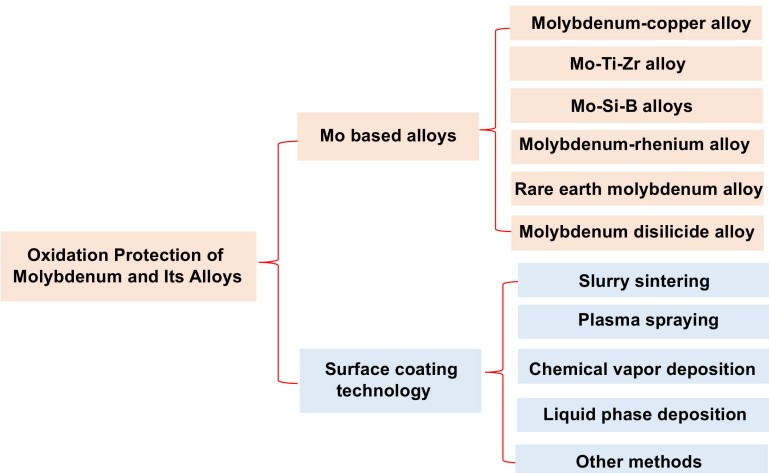

**Figure 1.** Overview of Mo alloy types and Mo and its alloy surface-coating technology.

## 2. Microstructure and Oxidation Behavior of Coatings

### 2.1. Coatings Prepared by Slurry Sintering (SS)

2.1.1. Microstructure and Growth Mechanism of SS Coatings

The slurry sintering (SS) method mixes alloy or silicide powder with binder in a certain proportion and then dissolves it in organic solvent to obtain the mixture. The mixture was evenly coated on the surface of the substrate, and then heated for a certain time in vacuum or Ar atmosphere, so that the substrate and mixture could be fully combined to form a coating on the surface [41,42]. As described in Table 1, the chemical composition and particle size of the mixture and process conditions have important effects on the composition, thickness, surface roughness and mechanical properties of silicide coatings [43–46]. Li et al. [43] investigated the influence of particle sizes of mixtures on surface grain sizes of the coatings. The reports show that the smaller the particle sizes of the mixture, the finer the grain size of the coatings, and they obtained coatings with surface grain size of only 1 to 5 μm by decreasing the particle size of the mixture. Similar results were reported by Wu et al. [44]. However, the surface roughness of the above two kinds of coating is still high. The surface roughness of most areas of the coating is above 15 μm, and the average roughness is 16.36 μm to 18.45 μm, as shown in Figure 2. The authors believe that the higher sintering temperature is the main reason for this result. It is worth noting that the interface layer of the two above coatings is relatively thin, only about 1 μm. In addition, the bonding strength and surface hardness of the coatings are not mentioned. Chakraborty et al. [45] successfully prepared a silicide coating on TZM surface with an interface thickness of 5 μm, a bonding strength of 25 MPa and a surface hardness of 2.00 GPa by slurry sintering technology, which has excellent mechanical properties. This is mainly due to the longer sintering time promoting the interdiffusion between the coating and the substrate, effectively. In addition, the surface quality of the coating will not decrease due to a sintering temperature that is too high.

**Table 1.** Summary of preparation process, coating composition and surface properties of TZM surface slurry coating.

| Substrate | Slurry Composition and Particle Size | | Process Conditions | | Coating Composition and Thickness (µm) | | Bond Strength (MPa) | Surface Hardness (GPa) | Grain Size (µm) | Refs. |
|---|---|---|---|---|---|---|---|---|---|---|
| | Composition (wt%) | Particle Size (µm) | Atmosphere | Treatment Time and Temperature | Outer Layer | Interface Layer | | | | |
| TZM | 75Si-10Mo-15Ti CN, EAC | 1.00–3.00 | Vacuum | 1450 °C, 15.00 min | $MoSi_2$-(Mo,Ti) $Si_2$ (120.00) | $(Mo,Ti)_5Si_3$ (1.00) | - | - | 1.00–5.00 | [43] |
| | 60Si-30Mo-10YSZ-$SiO_2$-PVB-$NH_4F$ | $1.00 \times 10^{-1}$ | Ar | 1450 °C, 1.00 h | $MoSi_2$-$ZrSi_2$-$SiO_2$ (120.00) | $Mo_5Si_3$ (1.00) | - | - | 2.00–5.00 | [44] |
| | MEK-PVB-10 to 20Si | 45.00 | Ar | 1200 °C, 2.00 h | $MoSi_2$ (60.00) | $Mo_5Si_3$ (5.00) | 25.00 | 2.00 | 10.00–20.00 | [45] |
| | 69.5Si-30Mo-0.5PVB-EA | - | Ar | 1450 °C, 1.00 h | $MoSi_2$ (96.00) | $Mo_5Si_3$ (3.00) | - | - | 2.00–4.00 | [46] |

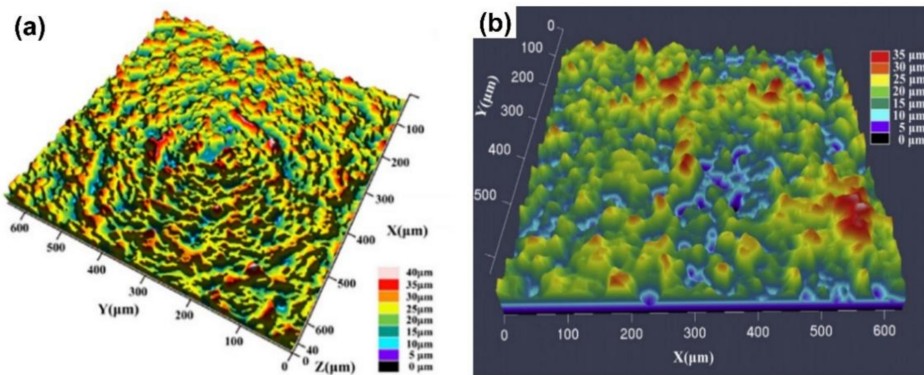

**Figure 2.** Roughness evolution of the Si-Mo coating (**a**) Reprinted with permission from [46]; reproduced from (Cai et al., 2017). surface roughness of the Si-Mo-5YSZ coating (**b**) Reprinted with permission from [44]; reproduced from (Cai et al., 2018).

The surface and corresponding cross-sectional images of the slurry coatings are shown in Figure 3. A great number of micro-cracks and holes are observed on the coating surface, and the volatilization of flux and binder during sintering is the main reasons for this result. Meanwhile, the high-temperature sintering shrinkage further aggravates the crack propagation and the increase of hole size [47], as shown in Figure 3a–d. However, the inner coatings are relatively dense and compact-bonded with the substrate. A thin interdiffusion zone (IDZ) can be clearly observed between the coating and the substrate. EDS analysis show that the atom ratio of Mo to Si is close to 5:3, which indicate that the inner coatings are a $Mo_5Si_3$ layer, as shown in Figure 3e,f. This is due to the decrease of diffusion rate with the decrease of silicon concentration during coating preparation, and finally a $Mo_5Si_3$ layer with lower Si concentration forms at the interface [48].

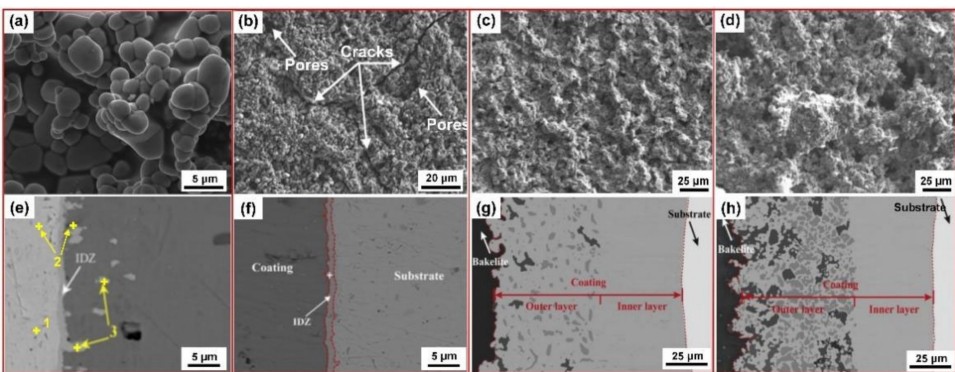

**Figure 3.** The images of surface and cross-sectional with different coating compositions. (**a**,**e**) $MoSi_2$/(Mo, Ti)$Si_2$ coating, Reprinted with permission from [43]; reproduced from (Li et al., 2018). (**b**,**f**) Si-Mo coating, Reprinted with permission from [46]; Reproduced from (Cai et al., 2017). (**c**,**g**) Si-Mo-5YSZ coating, (**d**,**h**) Si-Mo-10YSZ coating, Reprinted with permission from [44]; reproduced from (Cai et al., 2018).

The growth mechanism of the SS coatings and the main equations involved in the reaction process are shown in Figure 4. With the mutual diffusion between mixture and substrate, a dense $MoSi_2$ layer formed on the molybdenum alloys substrate. The content of Si element gradually decreased in the process of diffusion into substrate, and finally a thin interface layer ($Mo_5Si_3$ and $Mo_3Si$) formed with low silicon concentration between $MoSi_2$ and substrate. The results show that the growth mechanism of the interface layer in the $MoSi_2$ coating system is the same as that in $MoSi_2$/Mo diffusion couple, but the growth rate of $Mo_5Si_3$ is much higher than that of $Mo_3Si$ [49,50]. In addition, an appropriate amount of beneficial elements (M), such as Zr, Ti and Y are usually added to the slurry mixture to optimize coating structure [51–53].

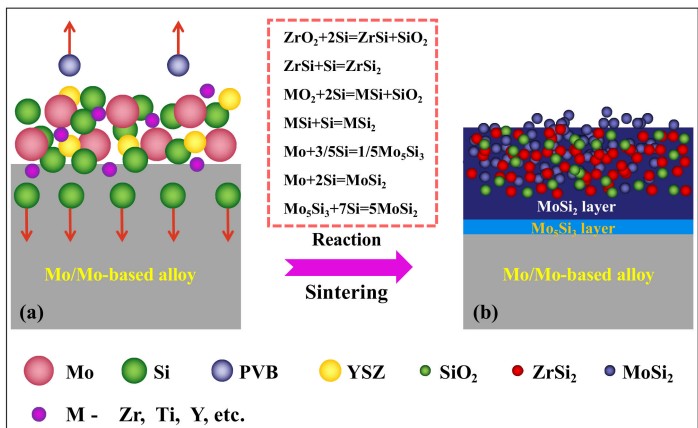

**Figure 4.** The diagram of growth mechanism of slurry sintering (SS) coating on molybdenum and Mo-based alloys. (**a**) Mixture composition and its diffusion law; (**b**) Structure of the coating after sintering reaction.

### 2.1.2. Oxidation Behavior and Mechanism of SS Coatings

The microstructure evolution and mass gain of the SS coatings before and after oxidation are shown in Table 2. It is observed that an oxide layer forms on the surface after oxidation, which is mainly composed of $SiO_2$, $TiO_2$, $Mo_5Si_3$, etc. Compared with the original coating, the thickness of the oxidized coating increases significantly, which is due to the volume of the coating expanding and the interface migration caused by the inter-diffusion reaction. However, the thickness of the $MoSi_2$ layer decreases significantly due to the growth of the oxide film and the migration of the interface layer. By contrast, the interdiffusion between the coating and the substrate becomes more sufficient with the increase of exposure time, resulting in a significant increase in the thickness of the interface layer dominated by $Mo_5Si_3$ [43–46]. The micro-structure and phase composition evolution during the high-temperature (above 1400 °C) oxidation is shown Figure 5. Relevant scholars believe that the integrity of the coating structure and the compactness of the oxide film are the key factors affecting the oxidation service life of the coating [48].

**Table 2.** Overview of the microstructure evolution and mass gain of SS coating on TZM alloy before and after oxidation.

| Substrate | Composition and Thickness of Coatings (μm) | | Exposure | Composition and Thickness of Oxidized Coatings (μm) | | | Mass Gain (mg·cm$^{-2}$) | Refs. |
|---|---|---|---|---|---|---|---|---|
| | Outer Layer | Interface Layer | | Oxide Layer | Intermediate Layer | Interface Layer | | |
| TZM | $MoSi_2$-(Mo, Ti) $Si_2$ (120.00) | $(Mo,Ti)_5Si_3$ (1.00) | 1600 °C, 5.00 h | $SiO_2$, $Mo_5Si_3$, $TiO_2$ (20.00–30.00) | $MoSi_2$-(Mo,Ti) $Si_2$ (70.00–75.00) | $Mo_5Si_3$ (53.00) | 4.00 | [43] |
| | $MoSi_2$-$ZrSi_2$-$SiO_2$ (120.00) | $Mo_5Si_3$ (1.00) | 1725 °C, 6.00 h | $SiO_2$, $ZrO_2$, $ZrSiO_4$ (77.00) | $MoSi_2$ (41.00) | $Mo_5Si_3$ (37.00) | 1.00 | [44] |
| | $MoSi_2$ (60.00) | $Mo_5Si_3$ (5.00) | 1000 °C, 5.00 h | - | - | - | Negligible | [45] |
| | $MoSi_2$ (96.00) | $Mo_5Si_3$ (3.00) | 1650 °C, 4.00 h | $SiO_2$ (24.00) | $MoSi_2$ (41.00) | $Mo_5Si_3$ (44.00) | $5.00 \times 10^{-1}$ | [46] |

The BSE images of the oxidized SS coatings are shown in Figure 6. The oxidation resistance of pure $MoSi_2$ coating is obviously poorer than that of composite coatings. We can clearly see from Figure 6a,d that the oxide layer of the pure $MoSi_2$ coating is very rough with a high porosity, and a large number of holes are observed. The addition of Ti can replace Mo atoms in the $MoSi_2$ coating, which changes the crystal structure of $MoSi_2$ and improves the coating density, as shown in Figure 6e. However, obvious cracks are still observed on the surface of the oxidized $MoSi_2$-(Mo,Ti)$Si_2$ composite coating [44]. This is due to the generation of $SiO_2$ partially crystallized as cristobalite during oxidation, and its

phase transformation is accompanied by a volume change, which reduces the adhesion of $SiO_2$ [47], as shown in Figure 6b. Wu et al. [46] report a Si-Mo-10YSZ coating with excellent oxidation performance at high temperature, and the mass gain after oxidation at 1725 °C for 6 h was only 1.00 mg·cm$^{-2}$. As Figure 6c reveals, $ZrO_2$ and $ZrSiO_4$ oxide particles are dispersed in the oxide film on the surface of the coating, which optimizes the structure of the oxide film and enhances its compactness.

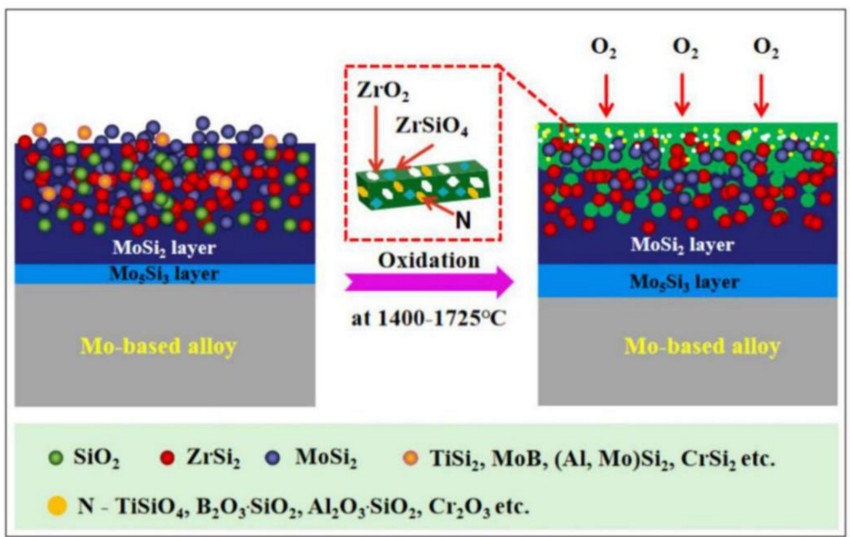

**Figure 5.** The diagram of oxidation mechanism of the SS coatings on molybdenum and its alloys.

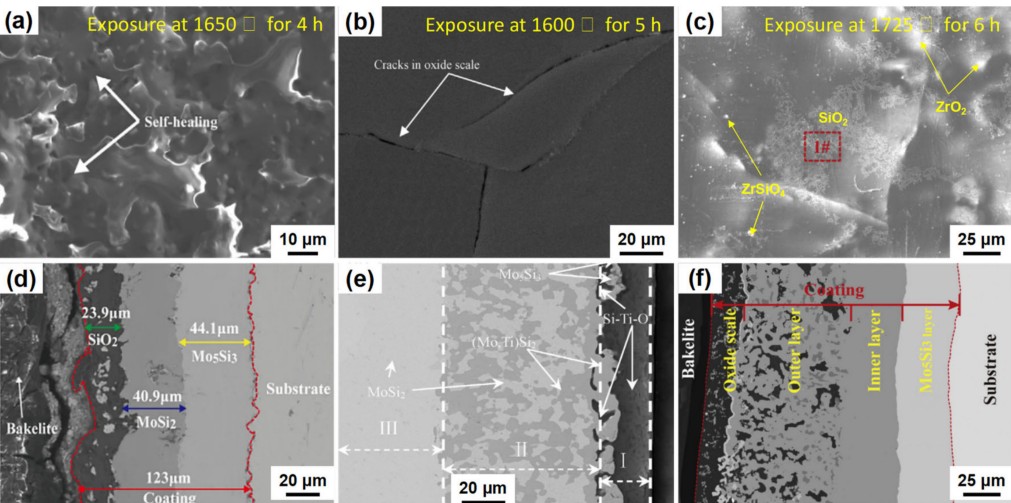

**Figure 6.** Surface and cross-sections images of SS coatings under different exposure conditions; (**a**,**d**) pure MoSi$_2$ coating, reprinted with permission from [45]; reproduced from (Chakraborty et al., 2016). (**b**,**e**) MoSi$_2$-(Mo, Ti)Si$_2$ coating, reprinted with permission from [46]; reproduced from (Cai et al., 2017). (**c**,**f**) Si-Mo-10YSZ coating, reprinted with permission from [44]; reproduced from (Cai et al., 2018).

### 2.2. Coatings Prepared by Plasma-Spraying Technique

### 2.2.1. Microstructure and Growth Mechanism of Plasma-Spraying Coatings

The plasma-spraying technique is one of the most widely used coating preparation in thermal-spraying technology. Its principle is heating and ionizing a certain gas ($N_2$, $H_2$, Ar, He or their mixture) by an electric arc. The generated high-energy plasma arc can heat powdery materials to molten or semi-molten state and spray them onto the substrate surface at high speed to form a coating [54–56]. Among them, the air plasma spraying

technique (APS), plasma-transferred arc (PTA) and spark plasma sintering (SPS) are widely used in the surface oxidation protection of Mo and its alloys. The growth mechanism and main reaction equations involved in the preparation of silicide coatings by the plasma-spraying technique are shown in Figure 7 [57–61]. Generally, the spraying material consists of Si, Mo, $MoSi_2$, MoB, $B_4C$, and $ZrO_2$, etc. The $MoSi_2$ particles are formed by silicon powder and molybdenum powder at high temperature, which are attached to the surface of the substrate. At high temperature, the Si element in $MoSi_2$ further diffuses into the substrate, and finally an interface layer dominated by $Mo_5Si_3$ is formed between the coating and the substrate [62,63].

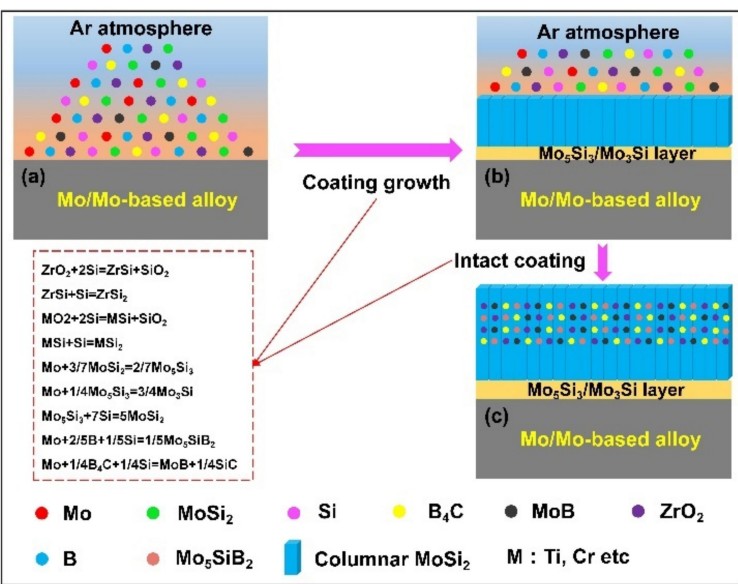

**Figure 7.** Diagram of the growth mechanism in plasma-spraying coating on molybdenum and its alloys. (**a**) Before spraying reaction (**b**) Coating structure at the initial stage of spraying (**c**) Structure of the coating after spraying.

Table 3 summarizes the process conditions and characteristics of antioxidant coatings prepared by the plasma-spraying technique. The microstructure and characteristics of coatings are mainly affected by powder feeding rate, spray gun power, vacuum degree, carrier gas flow rate, spraying distance, plasma gas composition [57–61]. Wang et al. [57] obtained a $MoSi_2$ coating on Mo substrate surface by the APS method, which had a high porosity and poor mechanical properties. The surface hardness, bonding strength, and porosity were 1.00 GPa, 10.00 MPa, and 28.73%, respectively. However, Deng et al. [58] prepared Mo–Si–B composite coating by the PTA process, and the surface hardness and porosity of the coating were 9.00 GPa and 18.00%, respectively. This was mainly due to the addition of element B, which improved the fluidity of Si and reduced the porosity of the coating. However, the coatings above have almost no interface layer, which results in very low bonding strength between the coating and the substrate. Chakraborty et al. [60] prepared a $MoSi_2/Mo_5Si_3$ gradient coating with a bonding strength of 40.00 MPa on TZM substrate by APS technology. A thicker interface layer formed between the coating and the substrate, which was due to the longer sintering time that makes the mutual diffusion between the coating and the substrate more sufficient. Baris et al. [61] obtained a $Mo_2BC/MoB$ coating on TZM with surface hardness of 21.00 GPa by SPS technology. A large number of fine granular boron and carbide dispersed phases were generated during sintering, which was the main reason for the increase of coating hardness.

Table 3. Summary of process, composition and properties of plasma spraying coatings on molybdenum and its alloys.

| Substrate | Spraying Material | Process Conditions | | | | | Composition and Thickness of Coatings (μm) | | Bond Strength (MPa) | Surface Hardness (GPa) | Porosity (%) | Refs. |
| | | Gas Flow (L·min$^{-1}$) | Powder (kW) | Distance (mm) | Treatment Temperature and Time | Pressure (MPa) | Outer Layer | Interface Layer | | | | |
| Mo | MoSi$_2$ | Ar: 40.00 H$_2$: 5.00 | 32.00 | 80.00 | - | - | MoSi$_2$, Mo$_5$Si$_3$ (600.00) | 0.00 | 10.00 | 1.00 | 29.00 | [57] |
| | Si, Mo, B | Ar: 6.00 | 47.00 | 20.00 | - | - | Mo$_3$Si- Mo$_5$Si$_3$- Mo$_5$SiB$_2$ (6000.00) | 0.00 | - | 9.00 | 18.00 | [58] |
| | MoSi$_2$ | - | - | - | 1500 °C, 5.00 min | 30.00 | MoSi$_2$ (500.00) | Mo$_5$Si$_3$ (20.00) | - | 10.00 | - | [59] |
| | MoSi$_2$, ZrO$_2$, MoB | - | - | - | 1500 °C, 5.00 min | 30.00 | MoSi$_2$, ZrO$_2$, MoB, Mo$_5$Si$_3$ (300.00) | Mo$_5$Si$_3$ (10.00) | - | 11.00 | - | |
| TZM | Si | - | 15.00 | 100.00 | 1100–1300 °C, 3.00 h | - | MoSi$_2$ (150.00) | Mo$_5$Si$_3$ (10.00) | 40.00 | 1.00 | - | [60] |
| | B$_4$C | - | - | - | 1420 °C, 10.00 min | 60.00 | Mo$_2$BC (214.00) | MoB (12.00) | - | 21.00 | - | [61] |

The SEM images and the process conditions of plasma-spraying coatings are shown in Figure 8 and Table 4, respectively [52]. It is obvious that the coatings have a very rough surface, with a large number of pores and a wide diameter. In addition, the coating surface contains a great deal of spherical particles, which is the result of "splashing" liquid droplets generated when the mist particles collide with the coating surface during the spraying process, then deposited on the coating surface again and cooled, as shown in Figure 8a–d. The densification of the cross-sectional coatings is very poor with many pores and uneven distribution, as shown in Figure 8e–h. The uneven particle size and melting degree of sprayed powder during spraying are the main reasons for this result. Meanwhile, the carrier gas remaining inside the coating also exacerbates this result [64,65]. The research shows that the microhardness and bonding strength of the coating can be improved by appropriately increasing the spray gun power or reducing the Ar gas flow rate [57].

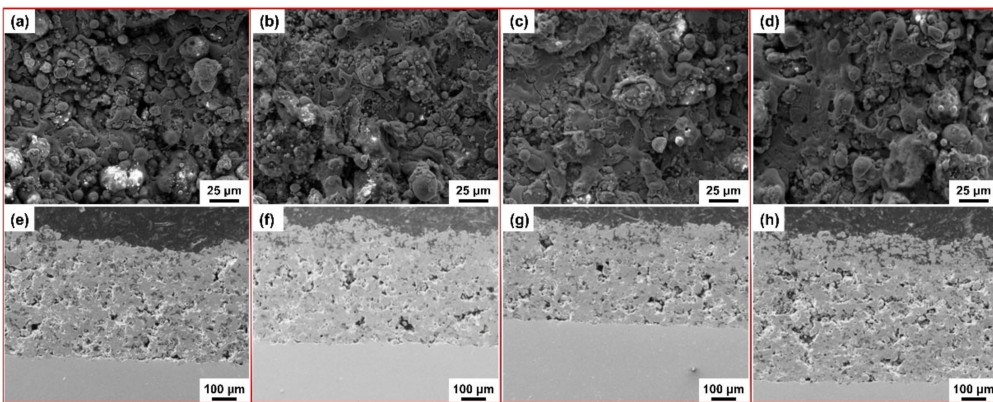

**Figure 8.** Surface (**a**–**d**) and cross-sectional (**e**–**h**) scanning electron microscopy (SEM) images of the $MoSi_2$ coatings on Mo with different process conditions. (**a**–**d**) represent MSi-1 to MSi-4, respectively. Reprinted with permission from [57]; reproduced from (Wang et al., 2013).

**Table 4.** Plasma-spraying conditions and the results of the characterization.

| Sample No. | Power (kW) | Primary Gas (Ar) Flow (L/min) | Second Gas ($H_2$) Flow (L/min) | Powder Feed Rate ($g \cdot min^{-1}$) | Distance (mm) | Hardness ($HV_{50}$) | Porosity (%) |
|---|---|---|---|---|---|---|---|
| MSi-1 | 30.00 | 40.00 | 5.00 | 32.00 | 120.00 | 1302.00 | 30.00 |
| MSi-2 | 30.00 | 50.00 | 5.00 | 32.00 | 120.00 | 1264.00 | 33.00 |
| MSi-3 | 32.00 | 40.00 | 5.00 | 32.00 | 120.00 | 1303.00 | 30.00 |
| MSi-4 | 32.00 | 50.00 | 5.00 | 32.00 | 120.00 | 1228.00 | 34.00 |

2.2.2. Oxidation Behavior and Mechanism of Plasma-Spraying Coatings

The micro-structure evolution and oxidation behavior of plasma-spraying coatings before and after oxidation are shown in Table 5. It should be noted that except for $Mo_2BC$ coating, the mass of the other coatings increases compared with that before oxidation. This is mainly due to the strong affinity force between C and oxygen. During oxidation, the volatilization rate of CO is greater than the formation rate of $B_2O_3$, resulting in the reduction of the overall quality of the coating. The oxidation mechanism and structural evolution of the plasma-spraying coatings is shown in Figure 9. The coating surface is gradually covered by a layer of $SiO_2$ with the volatilization of $MoO_3$. However, the strong oxidizing volatilization and volume expansion will further aggravate the surface defects. Finally, the coating fails due to the rapid consumption of the main part. The images of plasma-sprayed coatings after oxidation are shown in Figure 10 [57–59]. It can be seen that the surface of the oxidized coating is very smooth without obvious cracks and holes, which is mainly composed of $SiO_2$ protective film, as shown in Figure 10a–c. Meanwhile, the results of

Figure 10d–f show that the holes and cracks in the coating were filled by $SiO_2$ [57,58]. The 3MoSi$_2$-MoB-3ZrO$_2$ composite coating shows good oxidation resistance compared to pure MoSi$_2$ coating. The analysis shows that the oxidized composite coating has dense structure, uniform composition and obvious layered structure, as shown in Figure 10f [59]. The mass gain ($\Delta$ m/S) of the coating is only $4.00 \times 10^{-2}$ mg·cm$^{-2}$ after oxidizing at 1400 °C for 80 h. By contrast, the MoSi$_2$ coating has failed to oxidize under the same conditions. This is due to the fact that a crack-free oxide film composed of $ZrSiO_4$ and $SiO_2$ phases forms on the surface of the composite coating, which effectively prevents further diffusion of oxygen, as shown in Figure 10c [59]. In addition, during coating preparation, the process parameters, surface roughness and substrate temperature will also have certain effects on the oxidation performance of the coating [66–68].

**Table 5.** Micro-structure evolution and mass gain of spark plasma sintering (SPS) coatings on molybdenum and its alloys before and after oxidation.

| Substrate | Composition and Thickness of Coatings (µm) | | Exposure | Composition and Thickness of Oxidized Coatings (µm) | | | Mass Gain (mg·cm$^{-2}$) | Refs. |
|---|---|---|---|---|---|---|---|---|
| | Outer Layer | Interface Layer | | Oxide Layer | Intermediate Layer | Interface Layer | | |
| Mo | MoSi$_2$, Mo$_5$Si$_3$ (600.00) | - | 1200 °C, 25.00 h | SiO$_2$ | MoSi$_2$ (215.00) | Mo$_5$Si$_3$ (10.00) | 2.00 | [57] |
| | Mo$_3$Si, Mo$_5$Si$_3$, Mo$_5$SiB$_2$ (6000.00) | Moss, Mo$_3$Si, Mo$_5$SiB$_2$ (80.00) | 1300 °C, 30.00 h | SiO$_2$, B$_2$O$_3$, MoO$_2$ (30.00) | Moss, SiO$_2$ (15.00) | Mo$_3$Si,Mo$_5$SiB$_2$ | 8.00 | [58] |
| | MoSi$_2$ (500.00) | Mo$_5$Si$_3$ (20.00) | 1400 °C, 80.00 h | SiO$_2$ | MoSi$_2$ | Mo$_5$Si$_3$ | Failure | [59] |
| | MoSi$_2$, ZrO$_2$, MoB, Mo$_5$Si$_3$ (300.00) | Mo$_5$Si$_3$ (10.00) | 1400 °C, 80.00 h | SiO$_2$, ZrSiO$_4$ (2.00) | MoSi$_2$, ZrO$_2$ (396.00) | Mo$_5$Si$_3$, MoB (88.00) | $4.00 \times 10^{-2}$ | |
| TZM | MoSi$_2$ (150.00) | Mo$_5$Si$_3$ (10.00) | 1000 °C, 50.00 h | SiO$_2$ (10.00) | MoSi$_2$ (100.00) | Mo$_5$Si$_3$ (68.00) | 1.00 | [60] |
| | Mo$_2$BC (214.00) | MoB, Mo$_2$B (12.00) | 1000 °C, 1.00 h | B$_2$O$_3$ (10.00) | Mo$_2$BC | MoB, Mo$_2$B | −12.00 | [61] |

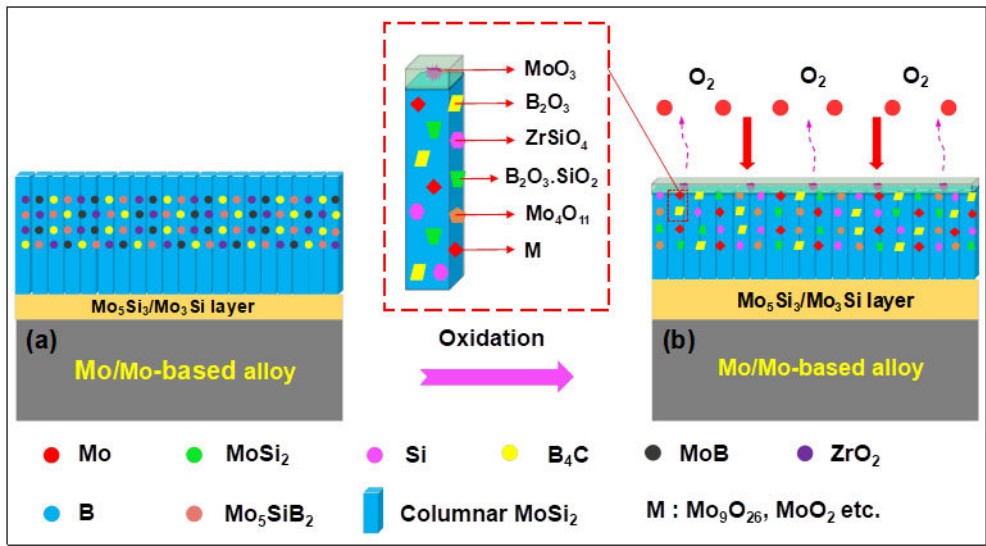

**Figure 9.** The diagram of oxidation mechanism of the plasma-spraying coatings on molybdenum and its alloys. (**a**) Coating structure before oxidation (**b**) Oxidized coating structure.

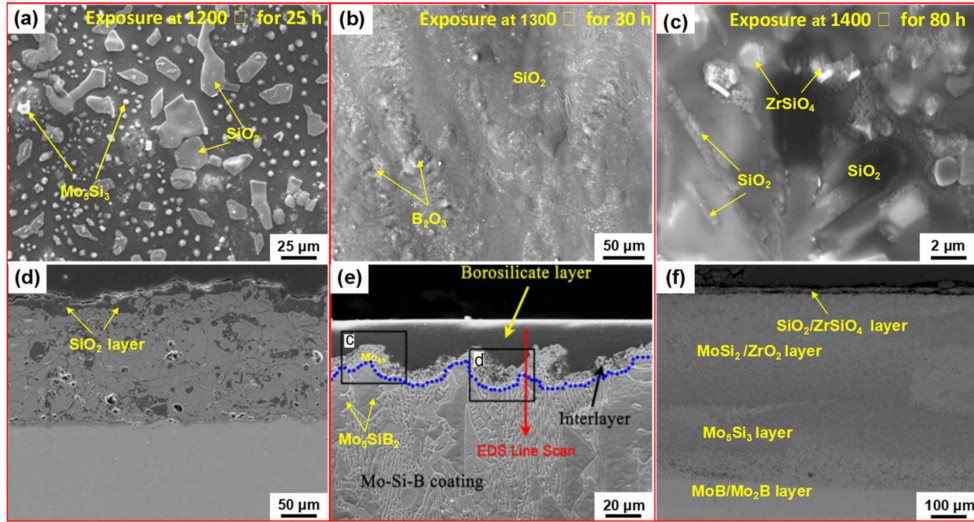

**Figure 10.** SEM image of surface and cross-section plasma spraying coatings after oxidation under different conditions. (**a**,**d**) Pure MoSi$_2$ coating, reprinted with permission from [57]; reproduced from (Deng et al., 2019). (**b**,**e**) Mo–Si–B coating, reprinted with permission from [58]; reproduced from (Zhu et al., 2019). (**c**,**f**) MoSi$_2$-MoB-ZrO$_2$ coating [59]. Reprinted with permission from [59]; reproduced from (Chakraborty et al., 2011).

### 2.3. Coatings Prepared by Chemical Vapor Deposition (CVD) Technology

2.3.1. Microstructure and Growth Mechanism of CVD Coatings

The principle behind chemical vapor deposition (CVD) technology is the process of using gaseous substances reacting with a solid substrate to generate solid deposits [69,70]. The process conditions and mechanical properties of the oxidation-resistant coatings prepared on molybdenum by the CVD technique as shown in Table 6 [71–75]. It is obvious that H$_2$ is often used as a carrier gas in the preparation of coatings. The images of the CVD coatings are shown in Figure 11 [71,72,75]. It can be seen that the coatings have a dense and homogeneous surface morphology with a granular structure, as shown in Figure 11a,b. However, the mismatch of thermal expansion coefficients (CTE) between MoSi$_2$ coating and Mo substrate makes obvious vertical cracks sprout inside the coating, as shown in Figure 11c,d. Huang et al. [73] prepared a TiB$_2$ coating on an Mo substrate with an average surface hardness of 28 GPa by CVD technology. However, the average surface hardness of MoSi$_2$ coating obtained under similar conditions is only 13 GPa. The author holds that the finer grains and lower surface roughness of the TiB$_2$ coating are the main reasons for this result.

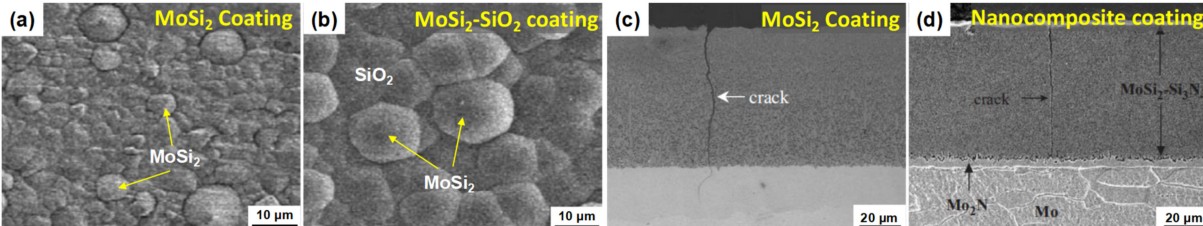

**Figure 11.** SEM images of different chemical vapor deposition (CVD) coatings on surface of Mo substrate. (**a**,**b**) Reprinted with permission from [71]; reproduced from (Nyutu et al., 2006). (**c**) Reprinted with permission from [72]; reproduced from (Yoon et al., 2005). (**d**) Reprinted with permission from [75]; reproduced from (Yoon et al., 2004).

**Table 6.** Summary of process, composition and properties of CVD coatings on Mo surface.

| Substrate | Composition of Gas Mixture | Process Conditions | | | Composition and Thickness of Coatings (μm) | | Bond Strength (MPa) | Hardness (GPa) | Surface Grain Size (μm) | Refs. |
|---|---|---|---|---|---|---|---|---|---|---|
| | | Gas Flow Rate (ml·min$^{-1}$) | Deposition Temperature (°C) | Deposition Time (h) | Outerlayer | Interface Layer | | | | |
| Mo | SiCl$_4$, H$_2$ | SiCl$_4$: 50.00 H$_2$: 100.00 | 620.00 | 3.00 | SiO$_2$ (3.00) | MoSi$_2$ (5.00) | - | - | 15.00 | [71] |
| | NH$_3$, SiCl$_4$, H$_2$ | NH$_3$: 100.00 H$_2$: 990.00 SiCl$_4$: 10.00 | 1100.00 | NH$_3$: 2.00 SiCl$_4$: 5.00 | MoSi$_2$, Si$_3$N$_4$ (72.00) | Mo$_2$N (5.00) | - | - | $3.00 \times 10^{-1}$ | [72] |
| | BCl$_3$, TiCl$_4$, H$_2$ | BCl$_3$: 195.00 TiCl$_4$: 130.00 H$_2$: 635.00 | 1000.00 | 2.00 | TiB$_2$ (13.00) | - | 7.00 | 28.00 | 2.00 | [73] |
| | WCl$_2$, H$_2$ | - | 1800.00 | 2.00 | W (160.00) | - | - | - | 20.00 | [74] |
| | CH$_4$, SiCl$_4$, H$_2$ | CH$_4$, H$_2$:200.00 SiCl$_4$: 10.00 H$_2$: 990.00 | 1200.00, 1100.00 | CH$_4$: 65.00 SiCl$_4$: 10.00 | SiC, MoSi$_2$ (60.00) | MO$_2$C (25.00) | - | - | $3.00 \times 10^{-1}$ | [75] |

　　　The growth mechanism of the oxidation-resistant protective coatings on molybdenum and its alloys prepared by CVD technology is summarized as shown in Figure 12. Si element decomposed from the mixed gas is deposited on substrate and reacts to generate a $MoSi_2$ coating at high temperature [71]. In order to enhance the low-temperature cyclic oxidation resistance of $MoSi_2$ coating, nitriding or carburizing treatment is usually carried out on the substrate before Si deposition. $NH_3$, $CH_4$, etc. are often used as nitrogen sources and carbon sources in this process to deposit on the substrate surface [76,77]. Then, a thinner $Mo_2N$ or $Mo_2C$ layer forms on the substrate surface, as shown in Figure 12b. The $Mo_2N$ or $Mo_2C$ layer will gradually consumed in the process of silicon deposition, most of them are replaced by a $MoSi_2$ layer with dispersed $Si_3N_4$ and SiC particles on the outer layer. The dispersed phase particles (such as $Si_3N_4$, SiC, etc.) can refine the grain size of $MoSi_2$, which significantly improves the mechanical properties and oxidation resistance of the coating, as shown in Figure 12c,d [72,77].

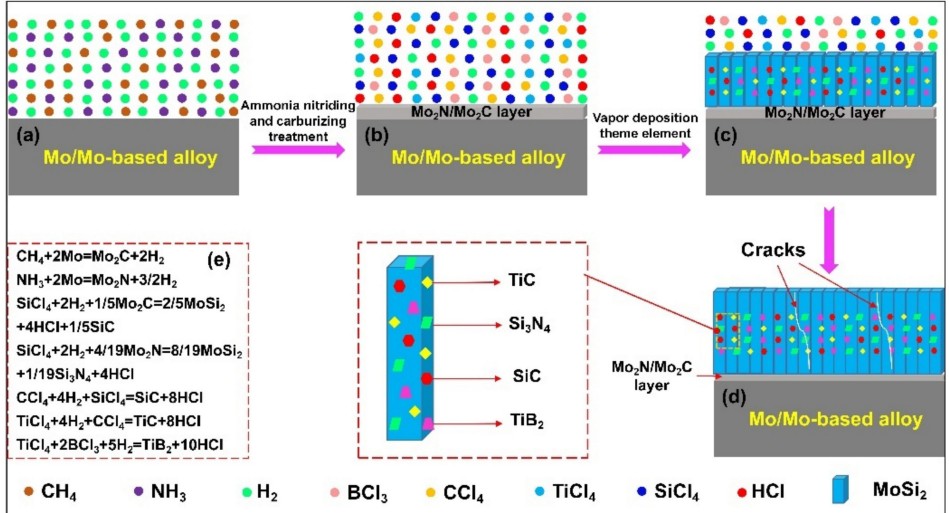

**Figure 12.** The diagram of growth mechanism of CVD coating on molybdenum and Mo-based alloys. (**a**–**d**) are the different stages of diffusion reaction, (**e**) is the equations involved in the reaction.

### 2.3.2. Oxidation Behavior and Mechanism of CVD Coatings

　　　Table 7 shows the microstructure evolution and mass gain of CVD coatings before and after oxidation under different conditions. Obviously, researchers mainly reported the oxidation of the coating at low temperature (500 °C to 1000 °C), and the oxidized coatings mainly consist of an oxide layer, intermediate layer and interface layer [71–75,78]. The images of CVD coatings after low-temperature cyclic oxidation are shown in Figure 13. At the initial oxidation stage, oxygen reacts violently with $MoSi_2$ at the crack and holes on the coating surface, and the generated granular $SiO_2$ spreads along the crack, as shown in Figure 13a. The coating surfaces are gradually covered by $SiO_2$ with the increase of cycle number. The thickness of oxide layer reaches 90 μm with a high porosity, as shown in Figure 13b,c. However, the oxidized $MoSi_2/β$-SiC nanocomposite coating remains intact, with only a small amount of $SiO_2$ attached to the coating surface. The oxide layer thickness is only 2 to 3 μm on average, as shown in Figure 13e,f respectively. This is mainly due to the preferential oxidation of SiC particles, which inhibits the oxidation of $MoSi_2$ and reduces the generation of volatile $MoO_3$. Meanwhile, CO generated during the oxidation process reduces the oxidation pressure in the system, which further reduces the oxidation rate [72]. The author believes that volume expansion caused by low-temperature oxidation causes the failure of the coating. Figure 14 shows the oxidation behavior and mechanism of CVD coating. It can be seen that the longitudinal cracks inside the coating further increase and expand due to the mismatch of thermal expansion coefficient between coating and substrate during the oxidation process [75]. In addition, Anton et al. Prepared Mo–Si thin film coating on the surface of Mo–Si–B alloy by magnetron sputtering. The oxidation test

shows that the coating can be used for 300 h at 1200 °C, while the inhibition time of medium temperature pulverization can reach 100 h at 800 °C. Moss- $Mo_3Si$-$Mo_5SiB_2$ phase is formed on the coating surface, which significantly improved its antioxidation activity [79–81].

**Table 7.** Microstructure evolution and mass gain of CVD coatings on molybdenum before and after oxidation.

| Substrate | Composition and Thickness of Coatings (μm) | | Exposure | Comments | Composition and Thickness of Oxidized Coatings (μm) | | Mass Gain (mg·cm$^{-2}$) | Refs. |
|---|---|---|---|---|---|---|---|---|
| | Outer Layer | Interface Layer | | | Oxide Layer | Intermediate Layer | | |
| Mo | $SiO_2$ (3.00) | $MoSi_2$ (5.00) | 1000 °C, 3.00 h | - | $SiO_2$, $MoO_3$ | $MoSi_2$-$Mo_5Si_3$ | 12.00 | [71] |
| | $MoSi_2$-$Si_3N_4$ (72.00) | $Mo_2N$ (5.00) | 500 °C, 1492.00 h | 1.00 h cycles | $Si_2ON_2$, $SiO_2$, $MoO_3Mo_4O_{11}$, $Mo_9O_{26}$, (3.00) | $MoSi_2$-$Si_3N_4$ (100.00) | $5.00 \times 10^{-1}$ | [72] |
| | $TiB_2$ (13.00) | - | 900 °C, 6.00 h | - | $TiO_2$, $B_2O_3$ | - | $8.00 \times 10^{-2}$ | [73] |
| | $TiB_2$ (13.00) | - | 450 °C, 5.00 h | - | $TiO_2$, $B_2O_3$ | - | $3.00 \times 10^{-2}$ | [78] |
| | W (160.00) | W/Mo (2.00) | - | - | - | - | - | [74] |
| | $MoSi_2$-SiC (60.00) | $MO_2C$ (25.00) | 500 °C, 1492.00 h | 1.00 h cycles | $SiO_2$, $MoO_3Mo_4O_{11}$, $Mo_9O_{26}$ (8.00) | $MoSi_2$-SiC (80.00) | $1.00 \times 10^{-2}$ | [75] |

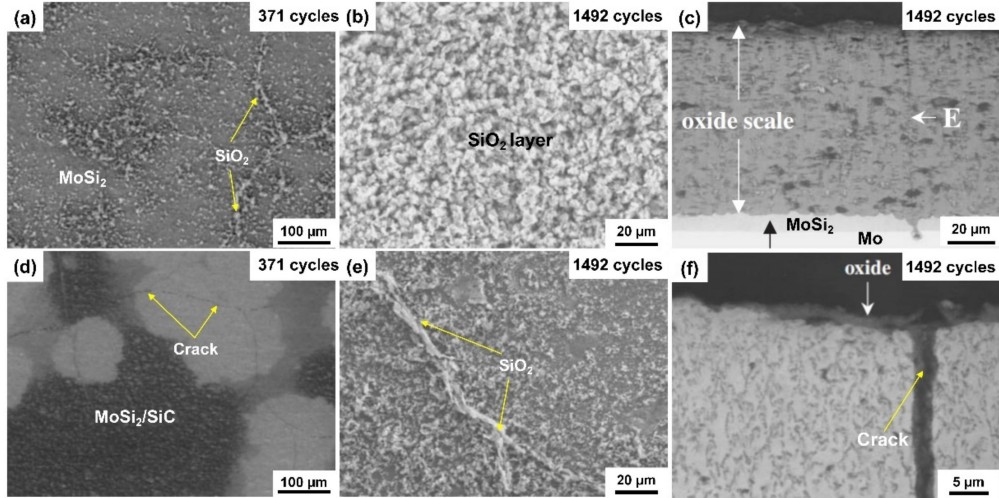

**Figure 13.** Surface and cross-sections images of oxidized coatings with different types and different cyclic oxidation times at 500 °C. (**a**–**c**) $MoSi_2$ coating; (**d**–**f**) $MoSi_2$/β-SiC nanocomposite coating. Reprinted with permission from [75]; reproduced from (Yoon et al., 2004).

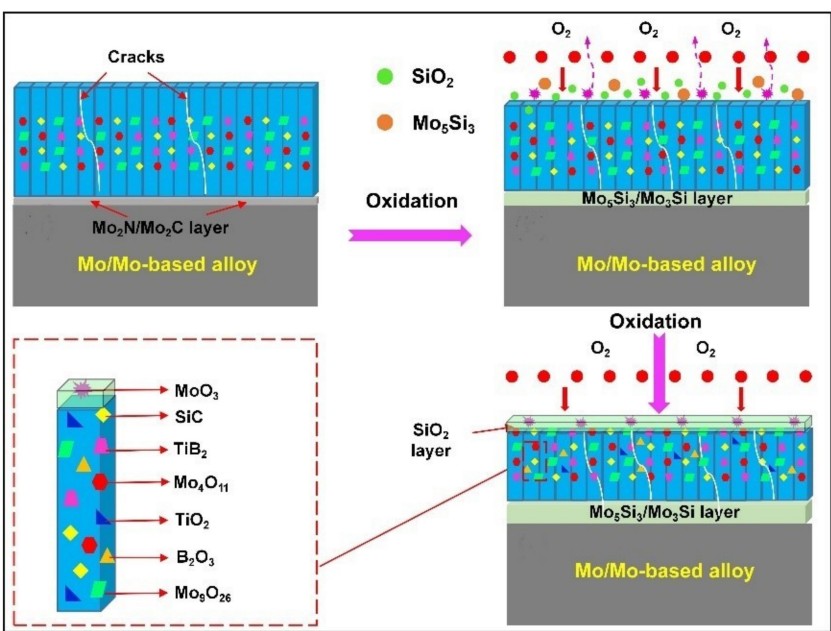

**Figure 14.** The mechanism diagram of oxidation of CVD coatings on Mo and Mo-based alloys.

*2.4. Coatings Prepared by LiquidPhase Deposition Technology*

2.4.1. Microstructure and Growth Mechanism of Liquid-Phase Deposition Coatings

Liquid-phase deposition technology inserts the refractory metal alloy into the alloy melt and prepares the intermetallic compound coating by a thermal diffusion reaction in a vacuum or inert gas atmosphere [82,83]. It is considered to be a promising surface coating technology for the oxidation protection of refractory metals [84]. The process details of liquid deposition technology for the preparation of Mo surface Si/Al coatings are shown in Table 8. It can be seen that hot dip time and temperature have important effects on coating composition, thickness and surface grain size [85–89]. Under high temperature and an Ar protection atmosphere, an intense diffusion reaction occurs between liquid silicon and substrate, and columnar $MoSi_2$ grains rapidly grow on the substrate's surface, as shown in Figure 15b. The thickness and surface grain size of the coating increase gradually with the increase of hot dip temperature and time. A thin interface layer ($Mo_5Si_3/Mo_3Si$ layer) with low silicon concentration is observed at the bottom of $MoSi_2$ grain, as shown in Figure 15c. Zhang et al. [90]. reported a Si-$MoSi_2$ coating on Mo substrate by the liquid deposition Si technology. The coating mainly consists of a $MoSi_2$ outer layer and $Mo_5Si_3/Mo_3Si$ interface layer, and the outer layer coating has a high surface silicon concentration and low roughness surface. It is worth noting that the surface of the coating has no cracks, holes and other defects. They find that the surface silicon concentration and grain size increase with the increase of deposition temperature and holding time, and the same conclusion has been reached by Wang et al. [89]. They report Al–Mo coatings with excellent oxidation resistance on the surface of Mo by the liquid-phase deposition Al technology. The Al–Mo coatings are mainly composed by out layer (Al-$Al_{12}Mo$ or Al-$Al_4Mo$ layer) and interface layer ($Al_8Mo_3$-$Al_4Mo$ or $Al_8Mo_3$ layer), and the interface layer is thicker than the outer layer. The SEM images of the liquid-phase deposition coatings are shown in Figure 16 [87]. The coatings surface are very smooth and dense, almost no cracks and holes are observed. In addition, high silicon concentration has been observed at $MoSi_2$ grain boundaries, as shown in Figure 16a–c. The cross-sectional morphology shows that the silicide coatings are composed of $MoSi_2$ columnar crystals with a thin transition layer ($Mo_5Si_3$ and $Mo_3Si$), as shown in Figure 16d–f.

**Table 8.** Overview of process, composition and surface properties of hot-dip coating on a molybdenum surface.

| Substrate | Osmotic Source and Purity | | Process Conditions | | Composition and Thickness of Coatings (μm) | | Si/Al Content on Coating Surface (wt%) | Coating Surface Grain Size (μm) | Refs. |
|---|---|---|---|---|---|---|---|---|---|
| | Infiltratesource | Purity (wt%) | Atmosphere | Hot Dip Temperatureand Time Min | Outer Layer | Interface Layer | | | |
| Mo | Si | 99.00 | Ar | 1460 °C, 20 min | Si-MoSi$_2$ (20.00) | Mo$_5$Si$_3$-Mo$_3$Si (3.00) | 47.00 | - | [85] |
| | | | | 1500 °C, 20 min | Si-MoSi$_2$ (22.00) | Mo$_5$Si$_3$-Mo$_3$Si (2.00) | - | - | [86] |
| | Si | 99.00 | | 1460 °C, 15 min | Si-MoSi$_2$ (15.00) | Mo$_5$Si$_3$-Mo$_3$Si (2.00) | 45.00 | 9.00 | [87] |
| | | | | 1520 °C, 15 min | Si-MoSi$_2$ (20.00) | Mo$_5$Si$_3$-Mo$_3$Si (4.00) | 56.00 | 7.00 | |
| | Si | 99.00 | | 1490 °C, 5 min | Si-MoSi$_2$ (12.00) | Mo$_5$Si$_3$-Mo$_3$Si (1.00) | 42.00 | 12.00 | [88] |
| | | | | 1490 °C, 15 min | Si-MoSi$_2$ (19.00) | Mo$_5$Si$_3$-Mo$_3$Si (4.00) | 56.00 | 7.00 | |
| | Al | 99.00 | No oxygen | 710 °C, 3 min | Al-Al$_{12}$Mo (30.00) | Al$_8$Mo$_3$-Al$_4$Mo (41.00) | 90.00 | 30.00 | [89] |
| | | | | 750 °C, 3 min | Al-Al$_4$Mo (35.00) | Al$_8$Mo$_3$ (51.00) | 81.00 | 55.00 | |

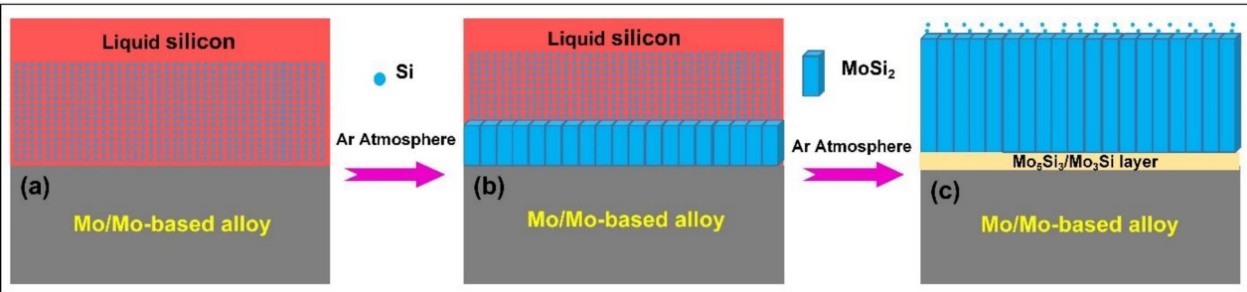

**Figure 15.** The growth mechanism diagram of liquid-phase deposition coating on molybdenum and its alloys. (**a**) is before reaction, (**b**) is the initial stage of the reaction, (**c**) is the late stage of reaction.

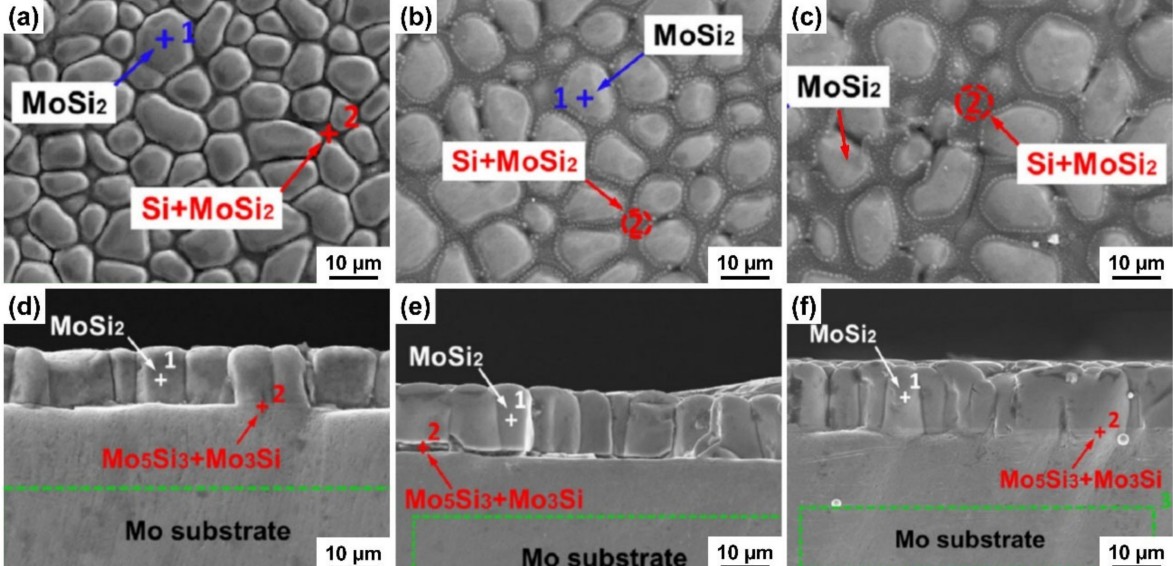

**Figure 16.** The surface and cross-section micrographs of Si-MoSi$_2$ functionally coatings obtained at different temperature for 15 min. 1460 °C (**a,d**), 1490 °C (**b,e**), 1520 °C (**c,f**). Reprinted with permission from [87]; reproduced from (Zhang et al., 2019).

In addition, Zhang et al. [91] also report a liquid deposition Si coating on TZM substrate. The results show that the surface roughness of the coating does not simply decrease with the increase of deposition time, as shown in Figure 17e–h. The surface of the sample deposited for 10 min is the roughest, and its Sa and Sq are 0.498 and 0.676 μm, respectively, as shown in Figure 17i. In addition, the surface roughness of the samples with deposition time of 15 min and 20 min is relatively low and close to each other, as shown in Figure 17j,k. However, their surface morphology are very different, as shown in Figure 17f,g. During the hot-dipping process, a great deal of molten silicon penetrated into the gap between the MoSi$_2$ particles, and it covered the grain surface during the cooling process. This gives the coatings obtained a smoother and more compact surface structure. It is worth noting that many banded textures were observed on the surface of the samples deposited for 25 min. This is due to the long deposition time reducing the Si viscosity, and the flow direction of the surface silicon changes during the extraction and cooling process of the sample, as shown in Figure 17d.

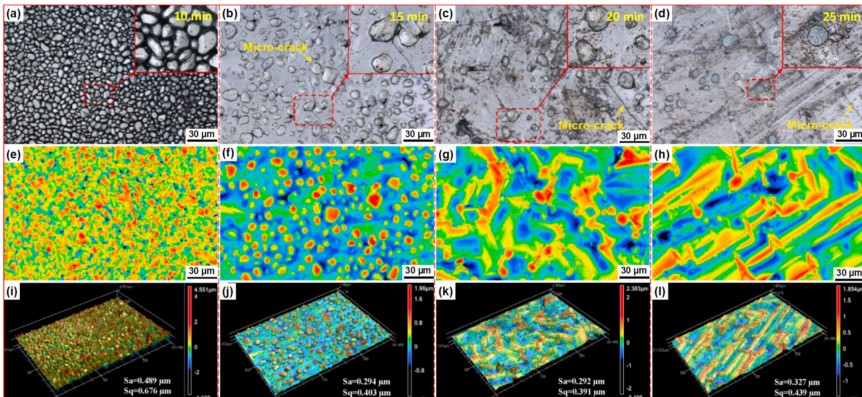

**Figure 17.** The CLSM (**a**–**d**), height distribution (**e**–**h**) and 3D images (**i**–**l**) of the coating obtained at 1480 °C for different times. Reprinted with permission from [91]; reproduced from (Zhang et al., 2021).

### 2.4.2. Oxidation Behavior and Mechanism of Liquid-Phase Deposition Coatings

The images of oxidized liquid-phase deposition coatings on Mo substrate are shown in Figure 18. After oxidation at 1200 °C for 2 h, the surface of Si-MoSi$_2$ coating is relatively rough with a small amount of pores, composed of SiO$_2$, Mo$_5$Si$_3$ and MoSi$_2$, as shown in Figure 18a [85]. However, a smooth and dense SiO$_2$ protective film forms on the coating surface when the oxidation temperature is 1600 °C, as shown in Figure 18d [89]. This is due to the good fluidity of SiO$_2$ at high temperature (above 1400 °C), which can fill the defects on the coating surface [90]. The oxidation mechanism and microstructure evolution of the coatings are shown in Figure 19. Compared with MoSi$_2$ phase, Si has a stronger affinity with oxygen in a high-temperature oxidation environment. Therefore, the silicon preferentially is oxidized to SiO$_2$, reducing the formation of volatile MoO$_3$. In order to improve the oxidation resistance of the Al–Mo coating, the coatings obtained at different hot dip temperatures are subjected to micro-arc oxidation (MAO) treatment (pre-oxidation). There are a lot of holes on the oxidized coating surface, which has typical MAO process characteristics, as shown in Figure 18b,c. The outer layer of the coatings is composed of Al$_2$O$_3$, the middle layer is an unoxidized aluminum dipping layer, and the inner layer is a diffusion layer of hot-dip aluminum, as shown in Figure 18e,f. The formation of the structure has great significance for delaying the diffusion of oxygen and prolonging the oxidation service life of the coating [88].

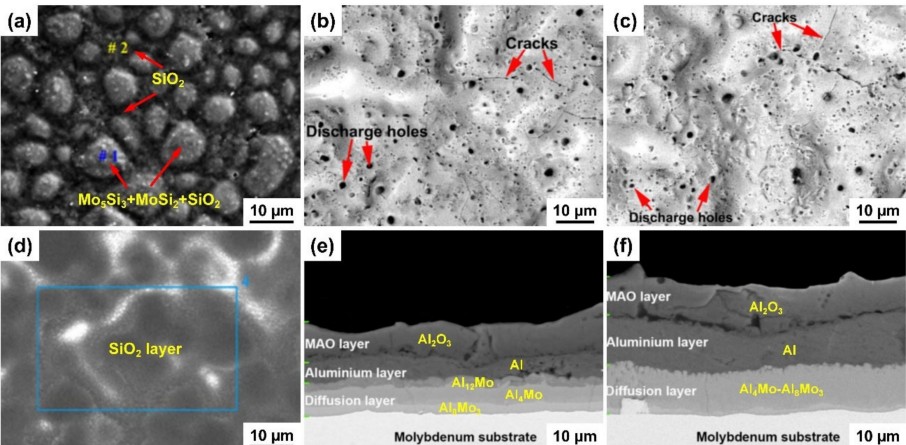

**Figure 18.** Surface topography of the oxidized Si-MoSi$_2$ coatings at different temperatures for 2 h, (**a**) 1200 °C, (**d**) 1600 °C. Reprinted with permission from [85]; reproduced from (Zhang et al., 2017). Images of Al–Mo coatings prepared at different temperature after MAO for 20 min; (**b**,**e**) 710 °C, (**c**,**f**) 750 °C. Reprinted with permission from [88]; reproduced from (Wang et al., 2020).

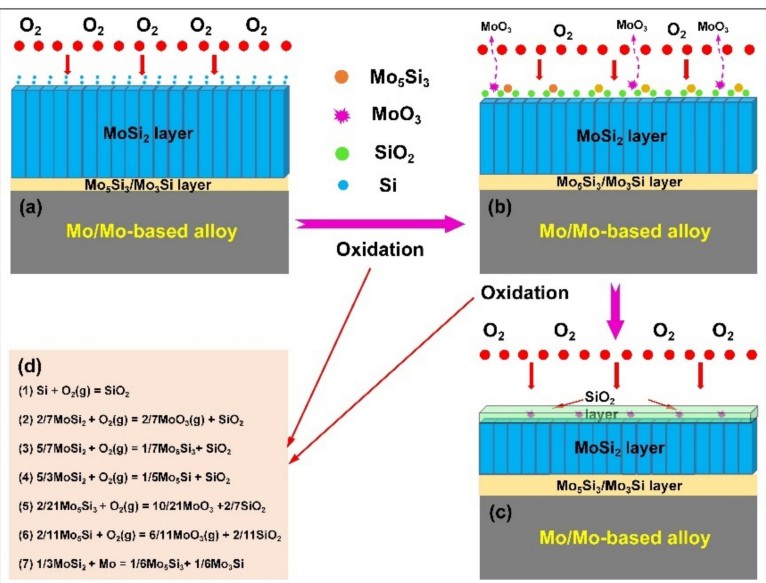

**Figure 19.** The diagram of oxidation mechanism of liquid-phase deposition coatings on Mo and its alloys. (**a**–**c**) are the coating structures before, during and after the oxidation reaction. (**d**) is the reaction equation involved in the oxidation process.

## 3. Conclusions and Prospects

As an important high-temperature structural material, the oxidation protection of Mo and its alloys has been of wide interest to relevant scholars. In this paper, the applications of various surface-coating preparation technologies in this field are reviewed, and the composition and oxidation characteristics of the coatings are shown in Figure 20. In addition, the characteristics of different coating preparation processes have also been analyzed and compared, and the details are shown in Table 9. During slurry sintering, due to volatilization of solvent and binder, the prepared coating has poor surface quality and high porosity. Reducing sintering temperature and prolonging sintering time can optimize coating structure and improve coating quality to a certain extent. The lower process temperature of CVD makes the preparation efficiency of the coating low and the preparation time long. However, the technology is suitable for workpieces with complex shapes and the coatings obtained have a good low temperature oxidation resistance. In contrast, plasma spraying and the hot-dip silicon method presented a high deposition efficiency due to high diffusion temperature. After 5 to 25 min of treatment, coatings several tens to several hundred microns thick can be obtained on the substrate surface. However, the plasma-spraying coatings have a high surface roughness and porosity because the spraying material is still mixed with a small quantity of residual gas and solid particles. It is worth noting that liquid-phase deposition coatings have a dense and smooth surface. This is conducive to the formation of protective oxide film on the coating surface in the oxidation process. However, the structure of the coatings are relatively simple, and the oxidation resistance of the coatings needs to be further studied. In addition, the molten salt method and laser-cladding technology have also been widely applied in the preparation of oxidation protective coatings on Mo and Mo-based alloys.

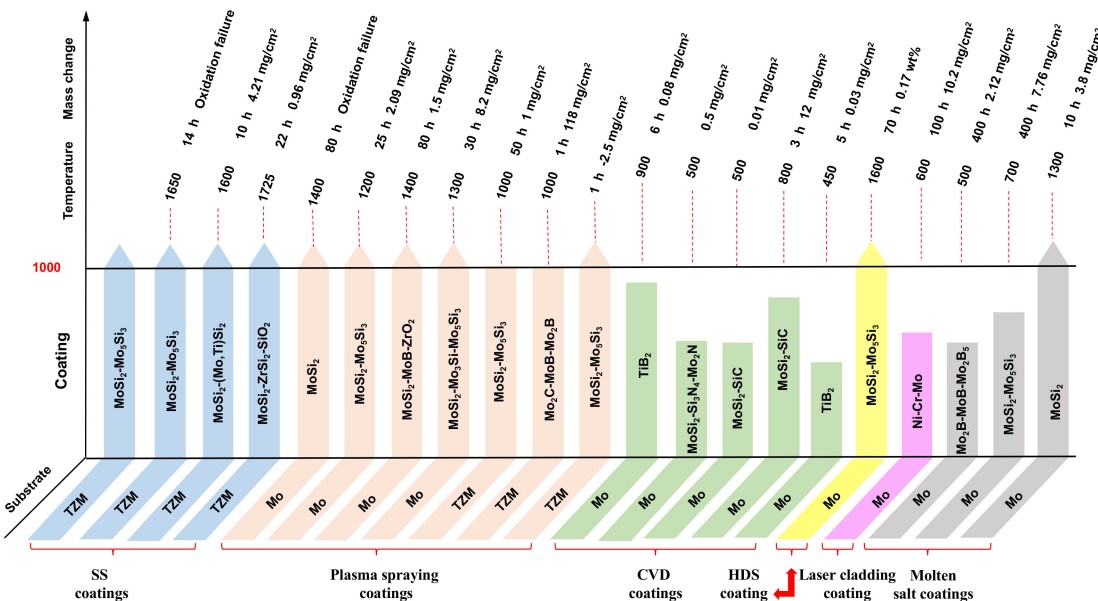

**Figure 20.** Overview of the composition and oxidation characteristics of silicide coating on molybdenum and its alloys.

**Table 9.** Summary of preparation methods and process characteristics of oxidation resistance coatings on molybdenum and its alloys.

| Method Category | Process Temperature | Time | Advantages | Disadvantages | Refs. |
|---|---|---|---|---|---|
| SS method | 1200–1450 °C | 15.00–120.00 min | 1. Simple preparation process, easy operation and low production cost. 2. That process adaptability is strong and the source materials are widely source. 3. That composition of the obtain coating is uniform. | 1. The surface quality of the coating is poor, and there are many cracks and holes on the surface of the coating. | [43–46] |
| SPS method | >10,000 °C | 5.00–10.00 min | 1. High spraying temperature 2. That operation is simple and the application range is wide. 3. That deposition rate is high, and the coat preparation cost is low | 1. That bond strength between the coating and the substrate is low. 2. High porosity of that coat | [57–61] |
| CVD method | 500–1000 °C | 2.00–10.00 h | 1. The application range is wide and is not limited by the shape of the substrate. 2. The coating composition has uniform thickness and good bonding with the substrate. | 1. The deposition temperature is low and the reaction time is long. | [71–75] |
| HD method | 1430–1560 °C | 10.00–25.00 min | 1. High hot dip temperature, short permeation time and high deposition efficiency. 2. The surface of the coating is smooth, the density is high, and the adhesion between the coating and the substrate is good. | 1. The structure of the coating is simple, and research on the oxidation resistance of the coating is relatively rare. | [85–91] |

The addition of appropriate beneficial elements in $MoSi_2$ coating has a great significance for improving the antioxidant properties of the coating. Ti element can replace Mo in $MoSi_2$ to form $TiSi_2$ solid solution, which improves the strength and hardness of the coating. In addition, a continuous and dense Si–Ti–O protective film is easy to form on the coating surface during the oxidation process. B element can not only reduce the viscosity of $SiO_2$

at high temperature (above 1400 °C) and improve the self-healing ability of the coating, it can also combine with Si element to form $Mo_5SiB_2$ with a lower diffusion coefficient and maintain the coating structure. N element is dispersed in the coating with a granular $Si_3N_4$, which improves its mechanical properties. In addition, $Si_3N_4$ distributed on the coating surface is preferentially oxidized, which reduces the oxygen partial pressure in the system and alleviates the oxidation of $MoSi_2$. Similarly, element C exists in the coating in the form of SiC and plays a similar role. With the addition of YSZ/Y, a continuous and dense $SiO_2$ protective film with $ZrO_2$ and $ZrSiO_4$ particles forms in the outer layer of the coating, which is conducive to stabilizing the oxide film structure and prolongs the oxidation service life of the coating.

Therefore, oxidation resistance and mechanical properties of the coating can be advanced in the following two ways. On the one hand, the preparation process of the coating should be optimized to ensure the coatings obtained have uniform compositions, compact structures and smooth surfaces. On the other hand, by introducing appropriate amounts of modified elements and the second phases, the structure of the coating can be optimized, the consumption of the coating can be slowed down, and the formation of the continuous and uniform protective oxide film formed on its surface can be accelerated. Furthermore, we can also organically combine the preparation process of a single coating to overcome the problems existing in its single application. This will be the future research and development direction in this field.

**Author Contributions:** The manuscript was written through contributions of all authors. Y.Z.: Conceptualization, Investigation, and Supervision. Y.Z. and T.F.: Writing original draft and image processing. T.F., K.C. and J.W.: Validation, Resources, Investigation, Writing—review and editing. X.Z., L.Y. and F.S.: Visualization, Writing—review and editing. All authors have read and agreed to the published version of the manuscript.

**Funding:** This work was supported by the Anhui Province Science Foundation for Excellent Young Scholars (2108085Y19) and the National Natural Science Foundation of China (No.51604049).

**Institutional Review Board Statement:** Not applicable for studies not involving humans or animals.

**Informed Consent Statement:** Not applicable.

**Data Availability Statement:** Not applicable.

**Conflicts of Interest:** The authors declare no conflict of interest.

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
