# Peer review of "Oxidation Protection of High-Temperature Coatings on the Surface of Mo-Based Alloys—A Review"

_coatings, doi:10.3390/coatings12020141_

Round 1

Reviewer 1 Report

The present review paper deals with the oxidation protection technique
especially coating. The review of different coating techniques is
presented.  The subject is useful for high temperature application.
Authors have presented the technical review in well mannered way. Before
it can be accepted for publication, authors are advised to add/modify
following
1.      The manuscript need to proofread for English language. There are many
sentences which does not make sense.
2.      Why pack cementation method is not included in this review, is not clear.
3.      Weight change plots of coated samples prepared by different coating
techniques at fixed temperature can be extremely useful for the readers
and easy to compare.
4.      Line number- 37- The alloying effect of several elements is highlighted
as inferior to surface coating without justification
5.      Line number 78 -- Any empirical relation between sintering temperature
vs bonding strength or surface roughness during slurry sintering is not
mentioned
6.      Line number 110-- Effect of Al, Cr and B in microstructure evolution
and mass gain during oxidation of coatings in slurry sintering technique
is unclear
7.      Line number 132--  Molydenum based alloys like TZM coatings behavior
and growth mechanism  by Chemical vapor deposition and hot dip method may
be highlighted and explained
8.      Line number 445-- Effect of combinatorial coating process on Mo/
Mo-based alloys and the effects can be included in the future research
parts
9.      A few more alloy systems such as solid solution strengthened Mo allay,
Mo-30W, newly developed Mo-TiSi-B and corresponding references and data
may be reviewed

Reviewer 2 Report

The article discusses the methods of surface protection of products from molybdenum alloys. In general, the article may be of some interest, but it needs fundamental correction - both from the point of view of the correctness of the English language, and from the point of view of numerous typos and inaccuracies.

The key note is a limited list of methods (for example, the PVD method is not considered, and this method can be effectively used).

The title of the article is not very clear: "Surface coating and oxidation protection ..." - but in fact, the oxide film is also a coating, and coatings are protection against oxidation.

Not very clear what "alloys-A review" means - a typo?

It makes no sense to introduce abbreviations in the Abstract, especially if they are no longer used in the Abstract.

Also, the constructions "In addition ..." "Finally ..." are not entirely appropriate for the Abstract.

It is a little strange to include such concepts as "Mechanism" and "Review" in the Keywords - how does this help the reader find what he needs?

The construction "... and so on " is not very appropriate for a scientific article.

"Fig. 1. shows" is an incorrect abbreviation at this position in the text.

"... and rare earth oxides etc are ..." - I recommend additional grammatical correction of the text, preferably with the participation of a native speaker.

Figure 1: There is no need to repeat the word "method" (moreover, the construction "CVD method" is not entirely correct - more correct is "CVD technology", while the most correct is simply CVD) and the word "alloy" every time, especially since we are talking about "alloys", not about "alloy" (as there may be different proportions of elements in the alloy).

"In the previous work, we have discussed the application of halide activated pack cementation (HAPC) in this field in detail [26]". - what does this have to do with the subject of this article?

This method is not even indicated in Fig. 1. How applicable is it for molybdenum alloys?

Why introduce an abbreviation?

"Wu [46] and Li [44] et al." - incorrect construction.

Figures 4 and 5 and 7 and 9 and 12 … and …: - original and previously unpublished, or the corresponding reference must be indicated.

"oxidation is shown Fig. 5." - Careful editing of the article is REQUIRED!

Table 3: it is incorrect to compare the thicknesses of 23.9 and 20-30 microns. Either in the first case, excessive accuracy, or in the second, insufficient. The horizontal lines necessary for understanding are missing in the table.

"The typical BSE images ..." - what does "typical" mean? Typical for what?

"plasma gas composition, etc [57-61]" is incorrect for a scientific article. What does "etc" mean? It does not matter? Why mention it then?

Table 4: There must be an identical number of decimal places in the compared parameters (0.73 and 21.4 - unacceptable!). Ditto for Table 5! The same for Table 6 (88.61 and 10) … Table 7 (0.092 - 0.3 and 1-2). And many other places - check it out!

"... rate of CO gas ..." - why do you need "gas"?

Table 10: Horizontal Separators Required!

Round 2

Reviewer 2 Report

In general, the authors took into account the recommendations and significantly improved the quality of the manuscript. However, I recommend making some more changes:

"oxidation protection of ... oxidation resistant coatings" - in my opinion, an obvious teutology.

And yet I find it useful to add horizontal lines in Tables 3, 4, 6, 7, 8 and 9  - without this, perception is difficult.

Table 5. - If 32.50, then it should be not 30, but 30.00. The same in other tables.

Author Response

Dear Reviewer:

Thanks very much for your comments concerning our manuscript entitled“Surface coating and oxidation protection for molybdenum and molybdenum-based alloys- A review” (ID: Coatings-1538164 ). Those comments are all valuable and very helpful for revising and improving our paper, as well as the important guiding significance to our researches. We have studied comments carefully and have made correction which we hope meet with approval. Revised portion are marked in red in the paper. The main corrections in the paper and the responds to the reviewers’ comments are as follows.

In general, the authors took into account the recommendations and significantly improved the quality of the manuscript. However, I recommend making some more changes:

  1. "oxidation protection of ... oxidation resistant coatings" - in my opinion, an obvious teutology.

Response: Thank you very much for your proposal. That's a great suggestion. The title of the article has been changed to "Oxidation protection of high temperature oxidation resistant coatings on the surface of Mo based alloys: A Review".

  1. And yet I find it useful to add horizontal lines in Tables 3, 4, 6, 7, 8 and 9 - without this, perception is difficult.

Response: Thank you very much for your proposal. The horizontal lines of above tables have been added.

  1. Table 5. - If 32.50, then it should be not 30, but 30.00. The same in other tables.

Response: Thank you very much for your proposal. The above problems in the article have been uniformly revised.